# The Role of Molecular Profiling in De-Escalation of Toxic Therapy in Breast Cancer

**DOI:** 10.3390/ijms26031332

**Published:** 2025-02-04

**Authors:** Sonia Y. Khan, Tonjeh Bah, Rakhshanda Layeequr Rahman

**Affiliations:** 1Department of Surgery, School of Medicine, University of Texas Rio Grande Valley, Edinburg, TX 78541, USA; sonia.khan@utrgv.edu; 2Department of Medicine, Case Western Reserve University School of Medicine, Cleveland, OH 44106, USA; tbah@metrohealth.org; 3MetroHealth Cancer Institute, MetroHealth, Cleveland, OH 44109, USA; 4Department of Surgery, Case Western Reserve University School of Medicine, Cleveland, OH 44106, USA

**Keywords:** breast cancer, de-escalation, precision medicine, molecular profiling

## Abstract

The prevalence and mortality associated with breast cancer have forced healthcare providers to leverage surgery, chemotherapy, radiation therapy, and immunotherapy to achieve a cure. Whereas mortality has significantly dropped over the decades, long-term toxicities and healthcare costs are prohibitive. Therefore, a better understanding of tumor biology through molecular profiling is being utilized for de-escalation of treatment where appropriate. As research evolves, there is growing evidence that less aggressive treatment regimens, when appropriately tailored, can be equally effective for certain patient populations. This approach not only enhances the quality of life for patients by reducing the financial, physical, and emotional burdens associated with more invasive therapies but also promotes a more personalized treatment strategy. By focusing on precision medicine and understanding the biological characteristics of tumors, healthcare providers and patients can make informed decisions that balance safety with efficacy. The field of molecular profiling is a promising avenue for precision-targeted de-escalation and escalation of therapy to minimize the risk–benefit ratio.

## 1. Introduction

Breast cancer therapy de-escalation treatment marks a transformation in oncology, focusing on lowering treatment intensity while still achieving effective outcomes for patients. Initially, radical mastectomy was the standard surgical approach, driven by the belief that aggressive removal of breast tissue and surrounding lymph nodes would improve survival rates. However, landmark trials such as the National Surgical Adjuvant Breast and Bowel Project (NSABP) B-06 trial initiated in the 1970s demonstrated that lumpectomy followed by radiation could provide equivalent outcomes, paving the way for more conservative surgical options [1,2]. This shift marked the beginning of de-escalation in surgical practices, as clinicians increasingly recognized that less invasive techniques could maintain efficacy while enhancing patients’ quality of life.

The introduction of genomic profiling has allowed oncologists to refine treatment approaches by identifying patients who may safely forgo chemo-, endocrine, or radiation therapy based on their low recurrence risk, representing significant de-escalation of systemic therapies. Collectively, these advancements highlight a transformative shift in breast cancer management, prioritizing personalized treatment strategies that balance efficacy with reduced treatment intensity. To create the optimal treatment for patients customized to their cancer, we must leverage the knowledge gained from genomic data when designing new trials to increase precision.

## 2. Progression of Evidence

### 2.1. De-Escalation of Surgery

There has been a growing shift toward the de-escalation of treatment modalities for early breast cancer to reduce the invasiveness of surgical interventions while maintaining effective treatment strategies (Figure 1).

Advancements in imaging and genetic profiling have led to a better understanding of tumor biology, allowing for curated approaches to treatment plans. There is a growing body of evidence supporting less aggressive surgical options, such as breast-conserving surgery, which can provide similar survival rates while improving the quality of life for patients.

Surgical treatment for breast cancer originated with extensive, aggressive procedures like the Halsted radical mastectomy where not only was all breast tissue removed but also the nipple–areola complex, pectoralis major and minor muscles, and level I-III axillary nodes [3]. This aggressive approach was questioned by Madden, who proposed significant modifications through conservation of chest-wall muscles [4], which was later reported to be equally effective in terms of survival [5]. However, as advancements in imaging and the effectiveness of chemo-endocrine and radiation therapies increased, the need for such drastic surgical treatment was called into question. The NSABP B-06 trial initiated in the 1970s served as a pivotal study first reported in 1985 [6] that moved the paradigm towards treatment de-escalation by establishing that lumpectomy followed by radiation therapy in women with early-stage breast cancer was just as effective as mastectomy, ushering in the era of breast-conservation treatments well established by long-term follow-up [1,2].

Now that the role of breast conservation is well established, and smaller cancers are detected with advances in screening, technological advancements in localization and induction of tumor necrosis have paved the way for forgoing surgical resection of breast cancer altogether [7]. The ICE3 trial documented the safety of forgoing surgical resection of clinical low-risk breast cancers in women older than 60 years of age presenting with invasive ductal carcinoma that could be targeted for cryoablation with ultrasound guidance [8]. Cryoablation both decreases the treatment and financial burden arising from surgical management while improving quality of life [9,10]. Ablative therapy trials have used clinical tumor profiles, such as tumor size, and phenotypes rather than molecular profile patient selection. Intuitively, the use of molecular data would enhance precision treatment for identifying candidates who could safely omit surgical resection of primary tumors.

Another target for surgical therapy in breast cancer has been the regional nodes (Figure 2).

Historic resections included surgical removal of internal mammary nodes in addition to the axillary nodes [11]. However, the 30-year follow-up of this approach showed no improvement in survival [12]. Whereas the internal mammary dissection was largely abandoned, axillary nodal dissection remained an important therapeutic approach. The goal of axillary lymph node dissections historically performed alongside mastectomies was for regional control and potentially increased survival. However, this regional control comes at the cost of potential nerve damage, seroma formation, and chronic lymphedema due to the disruption of the lymphatic drainage and anatomical proximity to brachial plexus branches. The NSABP B-04 trial found no significant difference in overall survival, disease-free survival, or distant disease recurrence in radical mastectomy, total mastectomy with axillary dissection if there were positive nodes, or total mastectomy with adjuvant radiation [13,14]. Then the NSABP-32 trial took one step further by establishing that sentinel lymph node resection alone could be a safe and effective alternative to axillary node dissection [15]. However, we must note that this comes with a 9.8% false negative rate while still carrying, albeit less frequent, side effects associated with axillary lymph node dissections [15]. While providing a route for de-escalation, we need to ensure that we are being judicious with how we determine the best treatment plans for optimal outcomes [16].

The American College of Surgeons Oncology Group Z0011 (ACOSOG Z0011) trial found patients with early-stage, clinically node-negative invasive primary breast cancer can forgo the more invasive axillary lymph node dissection for a sentinel lymph node dissection with whole-breast irradiation and adjuvant systemic therapy with similar survival outcomes in patients with 1–2 positive nodes [17]. This finding aligned with that of the International Breast Cancer Study Group Trial (IBCSG) 23–01 randomized control trial [18]. Alongside this, the After Mapping of the Axilla: Radiotherapy for Surgery (AMAROS) trial found noninferiority in axillary radiation therapy over axillary lymph node dissection in early-stage invasive breast cancer with 1–2 positive sentinel lymph nodes [19,20]. These trials collectively support the de-escalation of axillary lymph node dissection for sentinel lymph node biopsies with adjuvant radiation, further promoting de-escalation.

Current treatment strategies pursue de-escalation based on the phenotypic traits of a cancer. The Choosing Wisely campaign and the National Comprehensive Cancer Network (NCCN) version 6.2024 guidelines recommend against sentinel lymph node biopsies in women over 70 with early-stage, hormone receptor (HR)-positive, Herceptin receptor 2 (HER2)-negative breast cancers [21]. However, the NCCN guidelines go further by saying axillary de-escalation can be applied in patients with “favorable tumors”, yet they do not define this term. Assuming they are using the same traits as those in the Choosing Wisely campaign for patients > 70 years with HR-positive, HER2-negative breast cancers, this only describes phenotypic traits, leaving room for guideline supplementation regarding patients’ genomic data. This creates the potential for omitting axillary lymph node dissections in patients with low-risk molecular profiles and negative sentinel lymph node biopsies as our next step in surgical de-escalation.

The Multi-Institutional Neo-adjuvant Therapy MammaPrint Project I (MINT) trial utilized data harnessed from MammaPrint and BluePrint, molecular profiling tools, to stratify stage II-II invasive breast cancer patients that were post neoadjuvant chemotherapy as either low or high risk of developing distant metastasis [22]. Genomically high-risk tumors had a greater proportion of nodal downstaging with neoadjuvant chemotherapy, opening the potential for less invasive axillary lymph node surgery [22].

### 2.2. De-Escalation of Chemo-Endocrine Therapy

Systemic therapy for breast cancer has also undergone a significant transformation since its introduction (Figure 3).

The first effective chemotherapy regimen included cyclophosphamide + methotrexate + 5-fluorouracil (CMF), reported by Bonadonna et al. in 1976 [23]. This cytotoxic drug protocol, while effective, led to significant morbidity. The Milan group’s description of the use of anthracycline (doxorubicin) opened the door for exploring a shorter drug regimen to minimize toxicity. Four cycles of doxorubicin + cyclophosphamide (AC) was just as efficacious as six cycles of CMF and became the gold standard in the 1990s [24]. Several modifications to cytotoxic regimens, particularly with the discovery of taxanes, have been reported with significant success, once again at the cost of morbidity and quality of life concerns that ensue [25]. Over the decades, more targeted regimens based on cancer histologic and phenotypic characteristics emerged, improving both survival and recurrence rates. Another venue for systemic therapy in breast cancer has been endocrine treatment since the historic description of the efficacy of bilateral oophorectomy in patients with advanced breast cancers [26]. Since that time, several approaches to endocrine therapy including selective estrogen receptor modulators, aromatase inhibitors, and selective estrogen receptor degraders have been investigated for hormone receptor-positive breast cancer [27]. In 1998, a meta-analysis on tamoxifen showed significantly reduced recurrences in ER-positive and ER status-unknown patients [28]. The Arimidex, Tamoxifen, Alone or in Combination (ATAC) trial showed that the aromatase inhibitor anastrozole could be an effective alternative to tamoxifen for HR-positive pre-menopausal women with fewer adverse side effects [29]. On the other hand, studies like the MA.17 and the Breast International Group (BIG) 1–98 trials showed increased-duration combination endocrine therapies provided little benefit to post-menopausal hormone-receptor-positive patients [30,31].

However, systemic chemo-endocrine therapies have variable effectiveness even within the same subtype of cancer [32]. De-escalation studies based solely on receptor status are insufficient, necessitating a more nuanced approach with the incorporation of genomic data. While studies such as the Protocol for Herceptin as Adjuvant therapy with Reduced Exposure (PHARE) trial have attempted to decrease the treatment burden of systemic treatments, they ultimately highlight the limitation of analyzing phenotypic data alone [33]. This is where genomic data play a vital role in better tailoring treatment to the biological complexity of each patient’s cancer while minimizing associated toxicities of treatment.

Several trials have already begun to show the value of genomic profiling in decreasing the toxic burden of unnecessary systemic treatments. TAILORx takes the first steps in their study analyzing the benefit of adjuvant chemotherapy in HR-positive, HER2-negative, axillary node-negative breast cancer patients while taking their Oncotype DX recurrence score into consideration. Specifically, those given an intermediate score assigned endocrine therapy alone were non-inferior to those treated with chemo-endocrine therapy, except for those < 50 years old [34]. This dramatically improves patient quality of life and treatment burden without the cost of effective treatment.

The MINDACT trial evaluated the role of MammaPrint in guiding adjuvant chemotherapy treatment in women with early-stage breast cancer based on high or low-risk genomic status alongside clinical data. Importantly, this trial included node-positive disease which added clinical risk based on anatomical stage. This showed that even if these data were discordant with patients having high clinical but low genomic risk, some patients could safely avoid adjuvant chemotherapy [35].

A Clinical Trial RX for Positive Node, Endocrine Responsive Breast Cancer (RxPonder) is another clinical trial that utilized genomic testing to further narrow our scope of adjuvant chemotherapy, reporting benefits over adjuvant endocrine therapy alone in HR-positive, HER2-negative pre-menopausal patients with 1–3 positive nodes with low recurrence scores but not in post-menopausal patients of the same type. However, there is room for selectivity of endocrine therapy, as well. The Trans-aTTom trial found utilization of the Breast Cancer Index can allow the de-escalation of systemic endocrine therapy, identifying low-risk patients who will not benefit from long-term endocrine therapy [36]. Recently, an Ultra-Low Risk threshold for the 70-gene signature (MammaPrint) was developed to identify indolent disease with excellent prognosis from the STO-3 study [37]. The MammaPrint 70-gene analysis of tumor tissue from the IKA trial revealed women with UltraLow Risk genomic profile regardless of nodal status did not benefit from prolonged endocrine therapy [38].

### 2.3. De-Escalation of Radiation Therapy

Radiation therapy has played a key role in breast cancer treatment for over 100 years; however, the last 50 years have seen significant advancements owing to imaging and software development that allowed for modifications (Figure 4).

While initially utilized as a cauterizing agent, improved technological development allowed for better calibration and targeted treatment [39]. In a few short years, external beam radiation therapy (EBRT) became a standard adjuvant treatment for breast cancer, drastically reducing local recurrence rates [32]. Further advancements refined radiation therapy with hypo-fractionated regimens delivering focused, higher doses of radiation over shorter periods, reducing treatment burden while maintaining effectiveness [40,41]. Modern radiotherapy improves precision further with methods like intensity-modulated radiotherapy (IMRT) and image-guided radiotherapy (IGRT), minimizing toxicity to healthy tissues compared to previous techniques [42,43].

While radiotherapy has allowed for the de-escalation of surgical treatments, there is also a shift towards focusing on radiation treatments from a blanket treatment offered to those undergoing BCT to specific patient populations that would benefit without compromising overall outcomes. The START A and B trials showed decreased radiation dosing and length of treatment in early-stage breast cancer patients to be just as effective with improved side effects [44]. The TARGIT-IOT trial showed early-stage invasive ductal carcinoma patients could forgo EBRT altogether by way of targeted intraoperative radiotherapy given as a single dose during the time of lumpectomy [45]. However, the basis for patient selection was on size and clinical nodal status without consideration for either tumor phenotypic or molecular traits.

The ultimate de-escalation of radiotherapy is to provide irradiation only in those expected to have a considerable response. The CALGB 9343 [46] and PRIME II [47] trials both investigated the role of adjuvant breast radiotherapy in ER-positive early-stage breast cancer patients after BCT who were already receiving adjuvant endocrine therapy. Both studies found that while forgoing adjuvant radiotherapy did have a slightly increased locoregional recurrence rate (4% vs. 1% with adjuvant radiation), there was no significant survival benefit [46,47]. The Positive Sentinel Node: Adjuvant Therapy Alone Versus Adjuvant Therapy Plus Clearance or Axillary Radiotherapy (POSNAC) trial currently in progress takes the results of the Z0011 trial and seeks to determine if adjuvant axillary radiation treatment is beneficial in women with early-stage breast cancer patients over standard adjuvant therapy (albeit this is not clearly defined as noted to be variable depending on local recommendations). The LUMINA trial found that women 55 years and older with T1N0, grade 1 or 2, luminal A breast cancer receiving BCT with adjuvant endocrine therapy could forgo radiation treatment altogether with a low risk of local recurrence [48]. These trials once again base their population on their early-stage status or phenotype.

Recently the role of de-escalation of radiation has employed molecular profile for appropriate patient selection. The DCISionRT test raised brows positively when it introduced a risk calculator to predict radiation benefits in patients with DCIS. It quantified the risk for DCIS recurrence and predicted the benefit of radiation after lumpectomy with negative margins. In the PreludeDx study, DCISionRT showed a statistically significant benefit in radiation in patients with a high Decision Score and no benefit in patients with lower scores [49].

The ongoing De-Escalation of Breast Radiation (DEBRA) trial is looking at omitting radiation therapy after lumpectomy in low-risk, hormone receptor-positive, HER2-negative, early-stage breast cancer [50]. Stratification into low-risk recurrence score is based on MammaPrint/Blueprint or Oncotype score, and the benefit of omitting radiation will avoid overtreating and improve patient satisfaction and quality of life. Further delineating tumors into risk status based on molecular profiling may allow adjuvant radiotherapy after BCT To be omitted, regardless of age.

There are not many trials utilizing genomic data to their fullest potential, and this is especially seen within the field of radiation oncology. The DCISionRT PREDICT and Oncotype DX DCIS trials do show utility in gene biosignatures to guide adjuvant radiation treatment decisions in DCIS, ultimately allowing better-targeted treatments, however, there are still studies in process and further precision to be gained [51,52]. The NRG-BR007 is a currently ongoing phase III trial with the aim to de-escalate adjuvant radiation in stage I HR-positive, HER2-negative, and Oncotype DX recurrence score < 18 patients in those status post-BCT [50]. There is a significant interest in the development of prospective randomized trials deploying the use of genomic biomarkers, biomarkers of the immune system, including tumor-infiltrating lymphocytes to develop individualized radiation therapeutic options [53].

## 3. Molecular Profiling Options

Molecular profiling analyzes the genetic and molecular characteristics of tumors, providing additional data points to patients and clinicians when deciding upon treatment plans. It is important to note that phenotyping based on anatomical stage and immunohistochemistry or in situ hybridization-based biological profile is fairly uniform across all the studies mentioned in the above sections. However, genomic profiling studies do not uniformly classify these tumors because there are several commercially available genomic tests with some level of discordance [54,55]. Routine genomic testing can guide optimal therapy decisions to achieve a disease-free state without ineffective, toxic treatments based on predicted patient outcomes; however, the scientific community would need to reconcile the discordance between different commercially available genomic assays. The American Society of Clinical Oncology (ASCO) guidelines already support the use of molecular testing via Oncotype DX, MammaPrint, Breast Cancer Index, and EndoPredict to guide adjuvant endocrine and chemotherapy in select early-stage HR-positive, HER2-negative breast cancer patients with 0–3 positive nodes [56]. There are a wide range of molecular testing options available but below we focus on discussing the most commonly clinically used.

Oncotype DX is one such commercially available test that is used in patients with early-stage HR-positive, HER2-negative breast cancer. Oncotype DX was developed after identifying 250 candidate genes that were analyzed in a total of 447 patients from three separate studies, which eventually led to the 21-gene profile and an algorithm for calculating a recurrence score (RS) [57]. The 21 genes are divided into two groups: 16 are cancer-related, and 5 are reference genes that serve as internal controls. A mathematical algorithm was used to generate a recurrence score, which classifies patients as low-, intermediate-, or high-risk. The test analyzes these 21 genes via reverse-transcription polymerase chain reaction (RT-PCR) to create a “Recurrence Score” ranging from 0 to 100 to predict the likelihood of distant cancer recurrence after treatment. A lower score means the patient has a lower chance of recurrence and may not benefit from chemotherapy over hormone therapy, creating a less toxic treatment plan. Genes analyzed range from HR-related genes to cell proliferation genes and invasion and metastasis genes. Sixteen of the genes analyzed are cancer-related, while five function as references for baseline expression levels of target genes.

MammaPrint is another molecular test aimed at assessing the risk of recurrence in patients with early-stage breast cancer. The 70 genes that make up the MammaPrint genomic assay were discovered through an un-biased analysis utilizing artificial intelligence to sort the genes from the entire human genomic profile of a cohort of breast cancers collected and stored by the Netherlands Cancer Institute, from women who had undergone surgery, but had not received any systemic therapy for their cancers. Despite the apparent similarity in clinical and histologic profiles, long-term follow up of these women revealed two groups: those who remained disease-free and those who developed metastasis within 5 years. Without the effect of systemic therapy to alter the outcome, this cohort provided an insight into the true biology of metastatic potential [58]. MammaPrint analyzes 70 genes through microarray technology and focuses on tumor proliferation and differentiation. Classification is delineated in a binary format where patients are deemed low or high risk instead of a provided score.

Analyzing 50 genes, Prosigna is clinically available to assess recurrence risk in early-stage breast cancer. This particular test uses the PAM50 gene signature to classify tumors into intrinsic subtypes (such as Luminal A, Luminal B, HER2-enriched, and Basal-like), originally reported by Charles Perou, [59] and provides a 0–100 Risk of Recurrence (ROR) score for 10-year recurrence. This helps identify patients who can safely forgo chemotherapy and those who would benefit from more aggressive treatment [60].

Breast Cancer Index (BCI) is yet another commercially available gene expression assay for breast cancer recurrence, integrating genomic data with clinical factors. Focusing on predicting late distant recurrence, BCI highlights HR-positive, HER2-negative patients with the most to benefit from an extended period of endocrine therapy and those that will have little benefit [61].

EndoPredict is a gene expression assay geared towards the prediction of the risk of distant metastasis. EndoPredict was developed in a training cohort of 964 ER-positive and HER2-negative breast tumor samples [62]. The test is composed of a 12-gene molecular score that incorporates clinicopathological features (tumor size and nodal status). Eight genes were selected as relevant for therapeutic decision making including (i) proliferation-associated genes, and (ii) estrogen receptor signaling-associated genes. The EndoPredict signature also includes three RNA normalization genes and one DNA contamination control gene. The 12-gene molecular score is calculated by the weighted expression of the 8 target genes, as normalized by the 3 normalization genes, as measured in formalin-fixed paraffin-embedded (FFPE) breast tumor tissue. The 12-gene molecular score is then combined with clinicopathologic features including tumor size and lymph node status to produce the EndoPredict (EPclin) score. Patients with an EPclin score of ≤3.3 are classified as low risk and those with an EPClin score > 3.3 are classified as high risk. Patients deemed low risk are likely to not benefit from systemic therapy and may benefit from a more conservative approach.

FoundationOne CDx (F1CDx), on the other hand, is a comprehensive genomic profiling assay that identifies specific gene mutations within tumors across over 300 genes. With F1CDx, a tumor tissue sample is obtained and processed to extract DNA. The DNA then undergoes next-generation sequencing which is analyzed for potential actionable mutations that may respond to specific therapies. This can be used in various stages of cancer but is particularly beneficial for advanced, metastatic, recurrent, or unresectable tumors to help personalize therapy.

Guardant360 is a unique liquid biopsy that detects genetic mutations in circulating tumor DNA (ctDNA) from blood samples. This makes it less invasive and more convenient for patients. Guardant360 still analyzes over 80 genes associated with cancer but ctDNA allows for real-time monitoring of tumor dynamics. This test is geared towards advanced solid tumors with the potential to de-escalate the use of aggressive chemotherapy to a less intensive, less aggressive, and more precise treatment.

Molecular testing allows a deeper look into the unique traits of individual cancers. This knowledge plays a crucial role in advancing clinical practice and personalization of treatment strategies to each patient’s unique genetic profile. Molecular or genomic profiling not only provides an avenue to safely de-escalate from unnecessary toxic treatments; it potentially identifies specific targets for high-risk patients, some of which have already seen new drug developments.

## 4. Trials of the Future

### 4.1. Design

Designing clinical trials for toxic breast cancer treatment de-escalation involves strategic integration of clinical factors and genomic data to design a medical plan tailored to individual patient needs. The classical tumor-type-centered approach to treatment plans must shift towards gene-directed treatment. This new era of precision medicine necessitates new trial designs to best incorporate molecular profiling into clinical practice. Fountzilas et al. highlight four such innovative trial designs: basket trials, umbrella trials, master protocols, and N-of-1 trials [63].

Basket trials enroll patients with different cancer types that share a common genetic mutation into the same cohort, regardless of location. This allows researchers to evaluate the efficacy of targeted therapies based on specific biomarkers rather than generalized histology or anatomic location. This also allows for increased statistical power of a study as there is no population limitation based on cancer type. Data analysis and interpretation must be performed with the appropriate statistical design as varying responses may still be present within different cancer types despite having the same biomarkers [64,65].

On the other side of the precision medicine spectrum, umbrella trials focus on a single cancer type but analyze multiple targeted therapies based on various genetic mutations within that cancer. This allows for multiple parallel investigations of targeted treatment plans towards specific biomarkers within the cancer type in question, accelerating effective treatment identification and implementation. However, as this requires complex statistical analysis, more studies are needed on how to effectively conduct umbrella trials to maximize their potential [66].

Platform trials hold value in their adaptive nature, allowing for additional or removal of treatment arms based on interim results. A common, shared control group based on a prespecified interim analysis plan is still utilized to evaluate multiple intervention groups [67]. These three trial types together fall under the overarching master protocols trial design, each allowing the evaluation of multiple hypotheses under a single framework.

Arguably the ultimate personalized trial design is the N-of-1 trial. These are single-patient crossover trials designed to evaluate the most effective treatment strategy for a specific patent based on their unique molecular profile. This necessitates the temporal separation of trials for each patient. To most effectively do so, these trials are randomized and double-blinded over multiple periods. As expected, this can more easily play a role in patient burden and requires a deeper look into its sustainability and how to best support implementation [68].

Innovative trial designs come with challenges. In precision medicine, trial designs are often complex, necessitating extensive planning and coordination. Robust statistical methods are needed to cross-analyze multiple arms of a study while ensuring adequate power and minimizing errors. This also involves the analysis of large amounts of data, requiring effective collaboration between multidisciplinary teams. However, these are hurdles that once passed, can lead to molecular-level precision in patient care.

### 4.2. Implementation

The first step in creating these studies is a better understanding of the molecular evolution of breast cancer. This is where programs like BIG’s Aiming to Understand the Molecular Aberrations in Metastatic Breast Cancer (AURORA) step in. The AURORA program compares the molecular profiles of primary and metastatic tumors to elucidate tumor evolution and resistance. The program also directs patients toward precision medicine-based clinical trials based on their specific genomic data [69]. Trials such as the REal-world Clinical Application of molecular profiling in Solid Tumors (RECAST) trial are already showing improved progression-free and overall survival in patients treated with molecularly matched therapies [70]. The Full-genome expression Linked to clinical data to Evaluate breast cancer outcomes (FLEX) study also serves to create an extensive database from MammaPrint and BluePrint to expedite the discovery and development of molecular profiles [71].

Designing clinical trials for the de-escalation and escalation of breast cancer therapy must involve a combination of precision medicine and genomic profiling for eligibility so that the trial outcomes can be translated appropriately into clinical medicine. Precision medicine trials come with their own tribulations. Hayes et al. highlight several challenges in the widespread use of genomic testing in the clinical setting [72]. The complexity of cancer genomics, discordance among tests, and the current lack of robust clinical evidence make it difficult to incorporate precision medicine to its potential. New genomic discoveries must also go hand in hand with existing therapies, particularly targeted therapies. Until there is regular applicable clinical use, the cost of comprehensive genomic testing will be prohibitive as insurance providers are unlikely to fund a test that is not yet universally available.

### 4.3. Limitations of Genomic Profiling

Current limitations of molecular profiling include the lack of standardized interpretation methods for genomic tests and the high costs associated with these technologies restricting its accessibility. Additionally, some of the findings in genomic profiling studies pertained to specific patient populations, potentially limiting their applicability to a broader audience. For example, the results from RxPONDER revealed worse outcomes for black patients with a similar Oncotype score compared to other racial groups, thus limiting its generalizability [73,74].

### 4.4. Future Directions

A tool is limited in its success by its accessibility—both in terms of generalizable use and power. Future research studies should address newer approaches to design easily marketable and accessible molecular tools. In designing of these trials, patients should be randomized according to their molecular profile to determine the best suitable treatment. In this booming era of immunotherapeutics, it is worthwhile exploring the combination of the results from genomic profiling with the use of immunotherapies.

## 5. Conclusions

Precision medicine in breast cancer treatment is a growing movement in oncology, aiming to reduce the intensity of therapy while maintaining effective patient outcomes. Overtreatment can lead to unnecessary side effects and long-term complications. De-escalation of treatment originated in the analysis of patient phenotypic data deemed low risk—such as those with early-stage, hormone receptor-positive cancers. However, it has become clear that this is no longer enough. By identifying specific molecular profiles and their effective treatment strategies, physicians can tailor treatment plans that most effectively and efficiently treat a tumor without sacrificing a patient’s overall well-being. Shifting our focus to personalized medicine not only enhances a patient’s quality of life but alleviates the psychological and social burden associated with aggressive treatments. However, in order to do so, we must stop being penny-wise and pound-foolish and invest in a better understanding and application of genomic data.

## Figures and Tables

**Figure 1 ijms-26-01332-f001:**
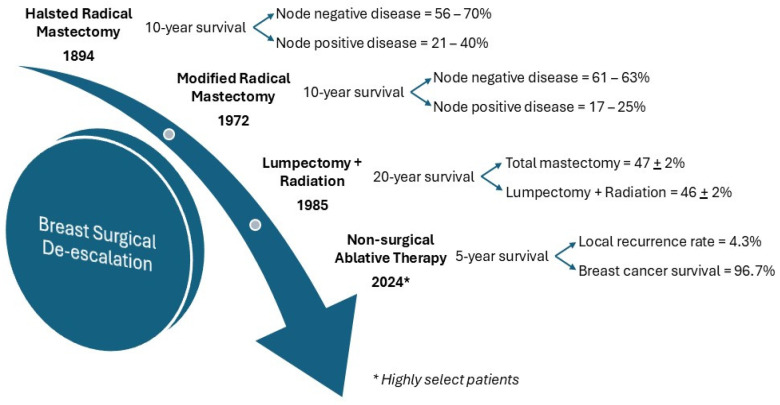
Timeline depiction of de-escalation for surgical therapy of the breast in breast cancer.

**Figure 2 ijms-26-01332-f002:**
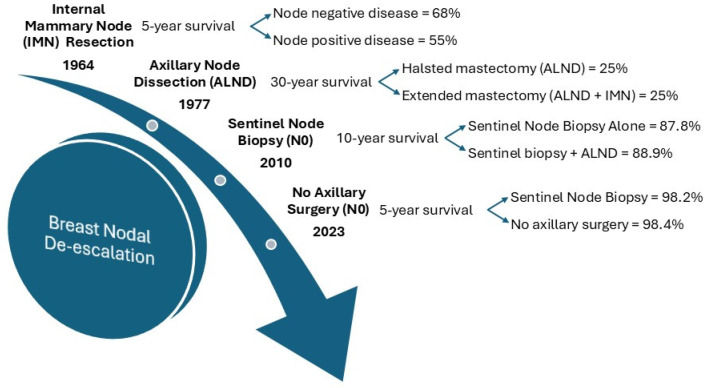
Timeline depiction of de-escalation for surgical approach to axilla in breast cancer.

**Figure 3 ijms-26-01332-f003:**
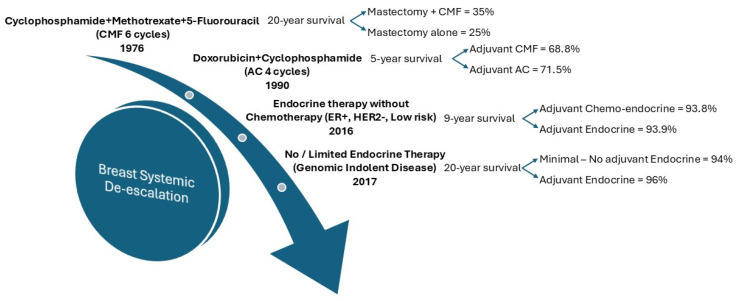
Timeline depiction of de-escalation for systemic therapy in breast cancer.

**Figure 4 ijms-26-01332-f004:**
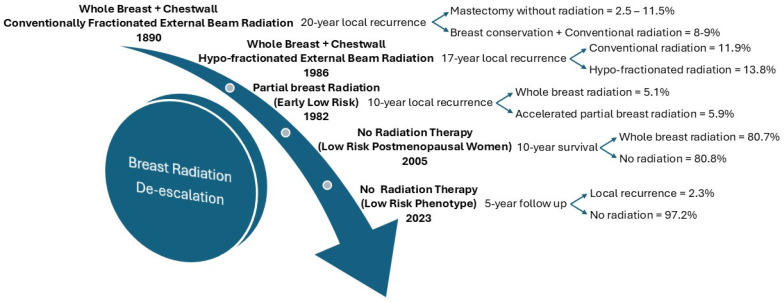
Timeline depiction of de-escalation for radiation therapy in breast cancer.

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
