# Peer review of "The Role of Molecular Profiling in De-Escalation of Toxic Therapy in Breast Cancer"

_ijms, 2025, doi:10.3390/ijms26031332_

Round 1
Reviewer 1 Report
Comments and Suggestions for Authors
The review article addresses the use of molecular analysis in the perspective of reducing toxic treatments for breast cancer. It talks about the potential of tailored therapy based on the molecular characteristics of tumors to reduce the intensity of treatment while preserving its effectiveness. It also reviews strategies such as less invasive surgical procedures, more precisely targeted radiation therapies, and chemo-endocrine treatment alternatives—all of these aimed at improving quality of life for patients. It also calls for novel clinical trial designs and individualized care based on information from molecular testing.
The article is important because it underscores the move towards personalized medicine and highlights the need for less harmful treatments. By understanding the molecular biology behind tumors, physicians can offer treatment options that are both safer and more effective. Not only does this approach improve prognosis, but it also reduces side effects and improves quality of life, making it a necessity for the future development of oncology care.
In my opinion, the article could be published if a few major issues were addressed:
1. The figures provided in the paper do not add much to the content. Indeed, they should be modified or substituted by more utilitarian figures, for example, showing the results of clinical trials or reduction in toxicity and improvement of the quality of life. Such alterations would have an effect of enhancing the usefulness of the figures due to their relevance with regard to the content they illustrate.
2. The information presented in some parts of the article is repeated, mainly on the point of molecular analysis. This may make the reader feel as though the ideas are repetitive. A more concise approach in expressing ideas clearly without much repetition would make the document more coherent and easier to understand.
3. The article's conclusions present a great overview; however, their effect may be enhanced by providing clear suggestions for future research studies. For example, suggesting new approaches to the design of molecular tools or the combination of these with immunotherapies would be beneficial. In this way, practical applications of the ideas presented in the article could be enhanced.
4. The title of the article contains the word "Title:", redundant.
5. References should meet the journal's needs.
Author Response
Please, see attached document with detailed response.

Reviewer 2 Report
Comments and Suggestions for Authors
The article titled "The Role of Molecular Profiling in De-escalation of Toxic Therapy in Breast Cancer", published in the International Journal of Molecular Sciences, explores the vital and timely topic of optimizing breast cancer treatment through therapy de-escalation. This approach aims to reduce the intensity of treatments while preserving their efficacy, with a focus on leveraging molecular profiling to enable personalized therapeutic strategies. By minimizing treatment toxicity, this method has the potential to significantly improve the quality of life for patients.
The authors present a comprehensive analysis of therapy de-escalation, discussing its application across surgery, chemotherapy, endocrine therapy, and radiotherapy. Their arguments are well-supported by numerous clinical studies, which enhance the credibility of the findings. A key strength of the article lies in its emphasis on personalized medicine, illustrating how tools such as Oncotype DX, MammaPrint, and EndoPredict enable more precise treatment decisions tailored to individual patient profiles. The content is structured in a clear and logical manner, making it accessible and easy to follow. Furthermore, the article introduces innovative research approaches, such as basket and umbrella trial designs, which highlight the growing integration of genomic data into clinical practice.
However, despite its strengths, the article has some limitations. It provides limited discussion on the potential challenges associated with molecular profiling, such as the lack of standardized interpretation methods for genomic tests and the high costs associated with these technologies. Additionally, while the article discusses important clinical trials, some of the findings pertain to specific patient populations, potentially limiting their applicability to a broader audience. The conclusions, though well-founded, could benefit from being more specific and should include a discussion of future research directions to provide greater depth.
Overall, the article offers a valuable contribution to the field of precision medicine by highlighting the role of molecular profiling in reducing treatment toxicity and improving patient outcomes in breast cancer therapy. However, future publications should address the practical barriers to implementing these technologies in everyday clinical practice and consider their impact on diverse patient groups. It is also recommended to include a reference to the publication DOI:10.3390/nu15112611, which explores the role of essential micronutrients in improving survival outcomes. Optimizing the levels of these micronutrients may enhance treatment responses and provide a broader perspective on factors influencing therapeutic success. Such an addition would strengthen the article by offering a more holistic view of patient care.
Author Response
please see attached document with detailed response.

Round 2
Reviewer 1 Report
Comments and Suggestions for Authors
The authors have revised the article in accordance with the suggestions provided. I recommend the publication of this manuscript.
Author Response
Thank you very much for your positive feedback